# Mixed Methods Studies on Breastfeeding: A Scoping Review

**DOI:** 10.3390/healthcare13070746

**Published:** 2025-03-27

**Authors:** Greyce Minarini, Eliane Lima, Karla Figueiredo, Nayara Pereira, Ana Paula Carmona, Mariana Bueno, Cândida Primo

**Affiliations:** 1Centro de Ciências da Saúde, Campus de Maruípe, Universidade Federal do Espírito Santo, Avenida Marechal Campos, 1.468, Vitória 29047-105, ES, Brazil; eliane.lima@ufes.br (E.L.); nayara_pereira97@hotmail.com (N.P.); candida.primo@ufes.br (C.P.); 2Nursing Department, Universidade Federal do Paraná, Av. Prefeito Lothário Meissner, 632, Bloco Didático II, Jardim Botânico, Curitiba 80210-170, PR, Brazil; karla.crozetafigueiredo@gmail.com; 3Nursing Research, Innovation, and Development Centre of Lisbon [CIDNUR], Escola Superior de Enfermagem de Lisboa, Avenida Professor Egas Moniz, 1600-190 Lisboa, Portugal; anapcarmona@esel.pt; 4Lawrence Bloomberg Faculty of Nursing, University of Toronto, 155 College Street, Suite 130, Toronto, ON M5T 1P8, Canada; mariana.bueno@utoronto.ca

**Keywords:** breastfeeding, maternal and child health centers, mixed methods, review

## Abstract

Prenatal and postnatal health guidance, along with women’s individual experiences, greatly influences mothers’ breastfeeding decisions and duration. To understand this better, a methodology combining quantitative and qualitative data is essential. **Objective:** To map the scientific production on the promotion, protection, and support of breastfeeding developed from mixed methods research. **Method:** A scoping review that followed the JBI and PRISMA-ScR methodology. The search was carried out in the following databases: Medline/PubMed, EMBASE, Cochrane, BVS, CINAHL, Web of Science, and BDTD. The guiding question was: Which studies on the promotion, protection, and support of breastfeeding were developed using mixed methods? The research took place from December 2023 to June 2024. **Results:** Of the 3569 studies found, 36 studies using mixed methods were included, with a prevalence of explanatory sequential approaches. Breastfeeding rates in the selected studies remain lower than global recommendations. The barriers to breastfeeding reported by mothers include biological; emotional; cultural; unfavorable social and hospital environment; difficulties in clinical management; lack of support from family, friends, health professionals, and employers; and skepticism about the benefits of breastfeeding. On the other hand, the main facilitators of breastfeeding reported were education and counseling during prenatal, childbirth, and postpartum periods; women’s intention to breastfeed; support from family, employer, spouse, and health professionals. **Conclusions:** The studies highlighted various issues related to breastfeeding rates, barriers, and facilitators across different contexts and cultures worldwide. The findings demonstrate that employing mixed methods enables a comprehensive and nuanced assessment of breastfeeding practices and the evaluation of relevant interventions.

## 1. Introduction

Breastfeeding plays a crucial role in preventing infections and chronic diseases due to its rich composition of antibodies, immunological factors, and essential nutrients. Studies indicate that breast milk protects against respiratory and gastrointestinal infections, as well as otitis media, thereby reducing infant morbidity and mortality associated with infectious diseases [1]. Moreover, breastfeeding is linked to a lower risk of chronic conditions such as obesity, type 2 diabetes, and hypertension due to its influence on metabolic regulation and immune system development. Therefore, promoting breastfeeding is a vital public health strategy that contributes to reducing the burden of diseases at various stages of life [2].

Mixed methods research can be an effective tool for analyzing complex health processes and systems, enabling researchers to address a variety of research questions that cannot be tackled by qualitative or quantitative methods alone [3]. When applied to the study of breastfeeding, mixed methods research has proven beneficial in helping scholars understand this complex, subjective, and experiential phenomenon. Recent studies show that integrating qualitative and quantitative data captures the intricacies of women’s experiences, including the emotional and social factors influencing their decision to breastfeed—insights that would be missed with isolated methods [4].

Furthermore, mixed methods research is particularly useful for examining the many factors that influence breastfeeding duration, such as biological, psychological, cultural, socioeconomic, and demographic aspects [5,6,7]. A study by Shobo and collaborators [7] highlighted that the interplay of these factors, along with women’s personal experiences, significantly affects the decision to initiate and continue breastfeeding. For example, prenatal and postnatal counseling provided by primary healthcare providers and hospitals directly influences a mother’s ability to start and maintain breastfeeding. Effective counseling, paired with social support, can significantly boost the rate of exclusive breastfeeding in the first six months, as demonstrated by a study conducted in the United States that combined qualitative and quantitative data [8].

Employing mixed methods in these studies offers a more comprehensive understanding of the barriers and facilitators to breastfeeding, providing a solid foundation for public policies and health interventions.

To understand all these factors and their influence on breastfeeding outcomes, it is necessary to work with a methodology that allows for a more comprehensive perception, incorporating both quantitative and qualitative data [9,10]. Mixed methods research (MMR) combines qualitative (QUAL) and quantitative (QUAN) research techniques to address a variety of complex phenomena more completely than either approach alone [11]. Therefore, it is necessary to conduct a broad search for objective and subjective elements that help to achieve a multifaceted understanding of the outcome of breastfeeding in different contexts and cultures, which can be achieved by applying mixed methods in research. Thus, the aim is to map the scientific production on the promotion, protection, and support of breastfeeding developed from mixed methods research.

## 2. Materials and Methods

This is a scoping review that followed the methodology of the JBI and the Preferred Reporting Items for Systematic Reviews and Meta-Analyses extension for Scoping Reviews (PRISMA-ScR) Checklist [12,13].

### 2.1. Protocol and Registration

The review protocol was registered on the Open Science Framework (OSF) platform, obtaining DOI 10.17605/OSF.IO/589Z2. The JBI Manual for Evidence Synthesis [12] was used to construct the review protocol.

### 2.2. Eligibility Criteria

After the search carried out on the OSF platform and the analysis of protocols of previous reviews, both completed and in progress, on the topic, no relevant studies were found. Based on this, the following research question was defined: “Which studies on the promotion, protection, and support of breastfeeding were developed using mixed methods?” The formulation of this question was based on the PCC mnemonic suggested by the JBI [11], which refers to: population (women), concept (breastfeeding, with an emphasis on care processes and practices, as well as the experiences of professionals and users), and context (health centers that promote and support breastfeeding, such as hospitals, maternity wards, and antenatal clinics).

Studies that focused on the promotion, protection, and support of breastfeeding using mixed methods were included in this review. Specifically, we considered studies that addressed the processes and practices of breastfeeding care, as well as the experiences of both health professionals and users in the context of breastfeeding. We also included studies conducted in various breastfeeding care settings, such as hospitals, maternity wards, and antenatal clinics. No restrictions were placed on publication date or language. Full-text publications that were freely accessible in journals available through the selected databases were retrieved.

### 2.3. Research Strategy

The research was conducted in three stages [11]. In the first stage, after the research question was developed, the DeCS/MeSH descriptors were identified and a bibliographic search was conducted in three databases: Google Scholar, Virtual Health Library (BVS), and National Library of Medicine (PubMed) in collaboration with a librarian. In the second stage, once the descriptors and keywords were defined in DeCS/MeSH, these were applied and combined in a search strategy adapted according to the specificities of the following databases: essential: Medline/PubMed, EMBASE (Elsevier, Amsterdã, The Netherlands), Cochrane; complementary sources: Lilacs/BVS, Cumulative Index to Nursing and Allied Health Literature (CINAHL-Ebsco, Birmingham, AL, EUA), SCOPUS (Elsevier), and Web of Science Core Collection (Clarivate Analytics, Philadelphia, PA, USA); and repositories: Brazilian Digital Library of Theses and Dissertations (BDTD), maintaining the similarities of combinations and using the Boolean operators AND and OR (Appendix A). The search took place in December 2023. Finally, the third stage includes checking the reference lists of the selected studies after reading the full text to find potential complementary studies relevant to address the research questions.

### 2.4. Studies’ Selection

Rayyan was used for studies selection. Two researchers independently (G.M.; N.P.) screened studies based on titles and abstracts. A third reviewer resolved conflicts on inclusion and exclusion (E.L.). As recommended by the JBI, the flowchart model in Figure 1 was used, detailing the studies’ selection process [13].

### 2.5. Data Extraction

For data extraction, a Word table was developed following the data extraction model provided by JBI [13], containing key information from the sources, such as author, year and country, objective, studies design, data collection instrument, theory, population and context (e.g., primary level, hospital), main results (quantitative and qualitative), limitations, and conclusion.

### 2.6. Summarizing and Reporting the Results

The content analysis technique by Bardin was used for the analysis and interpretation of the results. The results are presented both descriptively and through graphs and tables. The quality of the studies was evaluated using the Mixed Methods Appraisal Tool (MMAT), version 2018. This tool was used to assess the reliability and relevance of the designs in the selected studies. After careful examination of each study, they were evaluated based on five specific criteria for mixed methods research, in addition to two initial questions that apply to both qualitative and quantitative studies within the MMAT [14].

After reading in full, the mixed methods studies were evaluated using the MMAT, which has two general criteria, namely: Are there clear research questions? Do the data collected allow the research questions to be answered? Five specific criteria were: (1) Is there an adequate justification for using a mixed methods design to address the research question? (2) Are the different components of the studies effectively integrated to answer the research question? (3) Are the results of the integration of the qualitative and quantitative components interpreted appropriately? (4) Are divergences and inconsistencies between quantitative and qualitative results adequately addressed? (5) Do the different components of the studies meet the quality criteria of each tradition of the methods involved?

The information was then categorized into three groups: breastfeeding rates and duration, barriers to breastfeeding, and facilitators of breastfeeding.

## 3. Results

The search yielded 3569 studies. After excluding 450 duplicates and reviewing titles and abstracts, 3045 studies that did not meet the inclusion criteria were removed. After reading the remaining 74 studies in full, 36 studies met all the inclusion criteria, as shown in the flowchart following the PRISMA-ScR guidelines (Figure 1).

Mixed methods studies were conducted in 19 countries, with the majority conducted in the United States (*n* = 7; 19.4%) and published in 2020 (*n* = 8; 22.2%) (Table 1). Regarding data collection sites, 36.1% occurred in the community (*n* = 13), 22.2% in hospitals (*n* = 8), 16.7% in basic health units (*n* = 6), 13.9% in maternity wards (*n* = 5), and 11.1% in outpatient clinics (*n* = 4). Most studies included pregnant and postpartum women (*n* = 31; 86.1%), while health professionals participated in five studies (*n* = 5; 13.9%) (Table 1).

The most common approaches used were sequential explanatory (*n* = 18; 50%), convergent (*n* = 14; 38.9%), and sequential exploratory (*n* = 4; 11.1%). These studies aimed to clarify the institutional challenges that health professionals face regarding breastfeeding [15,16], utilizing a sequential explanatory approach. Furthermore, they explored mothers’ experiences and assessed the knowledge, attitudes, and practices of healthcare professionals related to breastfeeding counseling [17,18]. One study also examined the relationships between personal and environmental factors affecting exclusive breastfeeding [19]. Additionally, another study described the design and evaluation of a training program for women on breastfeeding, which also followed a sequential explanatory approach.

The QUAN and QUAL methods used were adequately reported individually in the included studies. The QUAL component of the studies used questionnaires or semi-structured interviews (*n* = 21, 58.3%), focus groups (*n* = 8, 22.2%), in-depth interviews (*n* = 5, 13.9%), observation methods (*n* = 1, 2.8%), and application of field diaries (*n* = 1, 2.8%). In the QUAN approach, 100% of the studies used questionnaires.

Most studies provided an adequate and complete description of the procedures followed in the sampling, data collection, and analysis stages of both components. On the other hand, the use of theories was present in the minority of the studies. The theory of planned behavior (*n* = 2, 5.6%) [19,20], grounded theory (*n* = 1, 2.8%) [21], breastfeeding self-efficacy theory (*n* = 1, 2.8%) [16,22], and Davis’s barrier analysis theory (*n* = 1, 2.8%) [23] were applied.

After reading and analyzing the thirty-six studies, it was found that only two studies [24,25] met all the criteria of the Mixed Methods Assessment Tool (MMAT) (Appendix A). Most studies included in this review lacked clarity regarding the theory used to support the analysis of the findings [4,5,6,15,17,25,26,27,28,29,30,31,32,33,34,35,36,37,38,39,40,41].

However, all studies included in the review were analyzed categorically. And, from the categorical analysis, common themes emerged that answered the following research question: “How are studies on the promotion, protection, and support of breastfeeding being developed using mixed methods?” These were organized into three categories: breastfeeding rates and duration, barriers, and facilitators, presented below.

### 3.1. Breastfeeding Rates and Duration

Although breastfeeding rates in the studies reviewed were often lower than the global recommendations set by the WHO, PAHO, and UNICEF, it is important to acknowledge the considerable variability across the different contexts analyzed. Factors such as varying study designs, specific population characteristics, and differences in how outcomes are measured can significantly influence these findings.

Breastfeeding in the first hour after birth was 49.4% in Mexico [42], 45% in Nigeria [27], and 39.7% in Indonesia [23]. Breastfeeding rates dropped within hours of birth, as observed in Nigeria, starting at 45% of mothers breastfeeding within the first hour of birth and decreasing to 29% within the first two hours [27]. In another study conducted in Nigeria, breastfeeding remained above 50% up to one hour after birth [7]. In Thailand, the cumulative proportions of mother–child pairs who initiated breastfeeding within two, three, and four hours after birth were 92.2%, 92.4%, and 94.7%, respectively [26]. These data indicate that there are significant variations in breastfeeding rates shortly after birth across different countries. However, a general trend shows that breastfeeding is more common in the first few hours after delivery, particularly in countries like Thailand, where the rates remain high. This suggests that cultural factors, public health policies, and postnatal care practices play an essential role in promoting the early initiation of breastfeeding.

Regarding exclusive breastfeeding rates at hospital discharge, in a single hospital between Thailand and Myanmar, 99.3% of full-term newborns and 98.8% of preterm newborns were discharged exclusively breastfeeding [26]. In Sweden, 82% of babies were exclusively breastfed at discharge [43] and in the studies by [37], this rate was higher than 90%. The data reveal significant variations in exclusive breastfeeding rates at hospital discharge across different countries. Thailand and Myanmar reported rates exceeding 98%, while Sweden’s rate was 82%. Other studies have indicated rates above 90%. These discrepancies may be attributed to hospital policies, breastfeeding support practices, and cultural factors. This highlights the necessity of developing tailored strategies to promote and sustain exclusive breastfeeding from birth.

The average length of exclusive breastfeeding observed in the United States was 5.8 months [41]. In Mexico, 44.8% of children were exclusively breastfed in the first month of life [42]. In South Africa, 34% of children were exclusively breastfed between four and eight weeks after birth [24]. On the other hand, in Tanzania, 76% of 316 postpartum women exclusively breastfed their children up to one month after birth [29]. Initially, 87.2% of Chinese and 75.6% of Irish mothers who gave birth in Ireland breastfed their children. The rates decreased to 49.1% at 3 months and 28.4% at 6 months for Chinese mothers, while the rate among Irish mothers at 6 months remained above 60% [44] (Figure 2).

Following recommendations from the WHO, UNICEF, and the Baby-Friendly Hospital Initiative to maintain exclusive breastfeeding rates at six months above 50%, several countries have assessed the duration of exclusive breastfeeding. Studies indicate that, at six months, exclusive breastfeeding rates are 19% in Nigeria [27], 20% in Haiti [28], 27% in Sweden [43], 37.4% in Thailand [19], and 35.3% in the USA [30] (Figure 2).

In China, women employed on formal contracts had a high rate of early initiation of breastfeeding, at 75.67%. However, this proportion dropped significantly to 12.65% at six months [32]. Another study found a similar decline, from 76% to 24.1%, over the same period [29]. In the USA, exclusive breastfeeding also declined over time: 33.4% of babies were exclusively breastfed up to three months, but only 16.9% continued to do so until one year of age [30]. At 12 months, breastfeeding rates varied between countries. While 17% of Chinese women and 7.6% of Irish women were still breastfeeding [44], rates were highest among Swedish women (21%) and significantly higher among American women (59%) [43,45].

The data show significant differences in the duration of exclusive breastfeeding between countries. These variations are influenced by cultural practices and local public health policies. This highlights the necessity for more effective and sustained strategies to promote exclusive breastfeeding, particularly for long-term goals, to achieve improved maternal and child health outcomes.

Regarding breastfeeding intention, a study conducted in Thailand found that all first-time pregnant women expressed the intention or experience of breastfeeding when questioned in a focus group [26]. Breastfeeding intention rates were mentioned by 99% of mothers in the studies conducted in the southern and central regions of Denmark [31] and 67% of women reported an intention to breastfeed for more than 5 months. A study conducted in the United States [39] indicates that 85.4% of women stated they intended to breastfeed.

When investigating the association between the breastfeeding practices of working mothers and their professional status and occupational areas related to agriculture, industry, and business, the intention to breastfeed was found in more than 90% [32]. Also, 94% of women had a high level of knowledge about breastfeeding, and 73% of mothers intended to breastfeed their babies for up to one year [27].

### 3.2. Barriers to Breastfeeding

The barriers to breastfeeding reported by mothers are biological; emotional; cultural; unfavorable social and hospital environments; difficulties in clinical management; lack of support from family, friends, health professionals, and employers; and skepticism about the benefits of breastfeeding.

#### 3.2.1. Biological and Physical Barriers

Scientific evidence has supported mothers’ reports of biological/physical barriers that influence breastfeeding, such as difficulty in latching on, sore/bruised and/or cracked nipples, change in the physical appearance of the breast associated with breastfeeding, insufficient milk production, mastitis, breast engorgement and nipple bleeding, which contributed to the woman’s decision to stop breastfeeding [15,21,31,35,38]. In Sweden, the findings on barriers to breastfeeding were represented by low milk production and insufficient breast milk to satisfy the baby when breastfeeding and/or gaining weight [43].

Although most mothers intend to breastfeed, many fear that they will not have enough milk for their baby and therefore prefer to supplement with formula, because they believe that donated milk may not be suitable for the baby and is not guaranteed to be safe and free from diseases, in addition to the belief that infant formula is recommended [23,42].

The child’s frequent crying and breastfeeding and short sleep periods were interpreted by mothers as signs of insufficient breast milk and drove the mothers’ decision to mixed feed their babies [24,29,42].

A mixed, explanatory sequential study, examining hospital practices, found that infants received infant formula during their hospital stay and that this was due to similar aspects, such as perception of insufficient milk, infant preferences, or cracked nipples. Other reasons included maternal need for rest and problems with breastfeeding technique [42]. Other barriers included twins, the baby’s refusal to breastfeed, being diabetic, smoking, or fear of breastfeeding because they were taking some medication [41].

#### 3.2.2. Lack of Support from Family and Friends

Mothers without support from family members and partners found it more difficult to breastfeed successfully [21]. The mother’s stress, the home environment, and the relationship with the child’s father are barriers and affect women’s mental health. According to them, these issues impair their ability to produce enough good-quality breast milk for their babies [24].

In the United Arab Emirates, women report difficulty in maintaining breastfeeding mainly due to employment, maternal exhaustion, and lack of family support [46].

The family environment and breastfeeding support significantly influence mothers’ decisions to start breastfeeding. Women considered the possibility of breastfeeding but felt discouraged by friends’ or family’s negative experiences with breastfeeding. African American and white adolescents reported that the lack of breastfeeding and negative experiences among their relatives and friends were barriers to the decision to breastfeed [20].

#### 3.2.3. Lack of Support from Healthcare Professionals

In Nigeria, barriers to breastfeeding related to support from health professionals were the inability of midwives to perform safe traditional practices with mothers, ineffective rooming-in practices, staff shortages, lack of privacy in the ward, and inadequate visiting hours policies. Furthermore, it was observed that pregnant women who were denied safe traditional birth practices, such as singing, praying, or reading religious books during labor, were five times more likely to not breastfeed their newborns in the first hour after birth compared to pregnant women who were allowed these practices [7].

Furthermore, despite contact with health and social service professionals during the first weeks postpartum, in most cases, adolescents did not receive support that would facilitate continued breastfeeding [20]. Corroborating this finding, less than half of health professionals reported asking parents how long they planned to breastfeed exclusively and how long they planned to continue breastfeeding after the introduction of complementary foods at the first or second visit. Although almost all providers reported asking parents about breastfeeding problems or concerns at the first or second visit, only half (53.6%) reported that they continued to ask about problems or concerns at the 2-month visit, and one third (34.3%) reported that they continued to ask about problems or concerns at the 4-month visit [17].

In Ireland, breastfeeding had to be stopped because difficulties were not addressed in a timely and appropriate manner. Women reported that they did not receive adequate support or advice from healthcare providers [44]. A lack of culturally appropriate breastfeeding promotion and a lack of breastfeeding education and encouragement from healthcare providers have been associated with breastfeeding failure [16,25,36].

#### 3.2.4. Skepticism About the Benefits of Breastfeeding

Skepticism about the benefits of breastfeeding, and the belief that children or other formula-fed individuals were healthy and intelligent, supported the idea that formula and breast milk are equivalent, which contributed to early weaning [21]. Concerns about the quality of human milk when a woman’s diet is unhealthy cause mothers to lose confidence in breast milk, and fears of harming the child may lead women to introduce formula milk and/or complementary foods at an early age [28].

Some mothers considered breastfeeding but were discouraged by grandmothers’ arguments against breast milk being sufficient, especially in the first six months after birth. In a study in south-western Nigeria, three of the grandmothers argued that introducing semi-solids or water alongside breast milk would allow the child to grow more quickly and enable their mothers to resume economic activities without a prolonged break [27].

The use of infant formula in Ireland was prevalent among participants, resulting in cessation or shorter duration of breastfeeding. This evidence is justified because, in this country, infant formula was widely considered to be of good quality, safe, and reasonably priced, compared to that produced in China [44].

Some Swedish mothers also explained that, when they started giving infant formula or solid foods, their breast milk dried up and it was difficult to continue breastfeeding [43].

#### 3.2.5. Unfavorable Social and Hospital Environments

The lack of space for breastfeeding in the workplace is a problem that hinders breastfeeding. Other barriers include the difficulty in balancing work and breastfeeding, the lack of funding to create a breastfeeding-friendly work environment, inadequate and periodic breaks to breastfeed or express milk, and the lack of adequate support for working mothers [6,15,16,21,25,41,43]. These barriers force mothers to adopt mixed method feeding practices to fulfill their professional and social responsibilities [32,46].

Social environments unfavorable to breastfeeding are another barrier. In the USA, most mothers reported that they did not want to breastfeed, and the most frequent reasons were: returning to school and feeling embarrassed about breastfeeding in public places [20].

Furthermore, the mother’s own will and external obstacles were additional reasons for stopping breastfeeding. An example of this reality is women with babies in the neonatal intensive care unit who have limited opportunities to be close to their babies [23].

### 3.3. Facilitators of Breastfeeding

The success of breastfeeding depends on numerous factors, including breastfeeding education and counseling that may occur during prenatal, childbirth, and postpartum care (Figure 3). Success is also influenced by the woman’s intention to breastfeed and support from her employer, family, and spouse.

Early breastfeeding education for newborns, during consultations in antenatal clinics and during the postpartum period, has been shown to facilitate breastfeeding problems, since the knowledge acquired by women during prenatal care influences their behavior after childbirth and can improve newborn survival [7].

Nurses play an important role in providing health education on breastfeeding during antenatal and postnatal visits at the health facility. The provision of targeted services by trained health professionals who provide breastfeeding support during antenatal and postnatal care has been associated with increases in breastfeeding rates, particularly in exclusive breastfeeding rates [27]. Another practice that encourages breastfeeding is the support offered by women in the community who have already breastfed, a service known as peer support [22]. Furthermore, women found prenatal consultations and postnatal contacts rewarding, enjoyable, encouraging, and valuable [23].

Counseling is a practice that encourages breastfeeding. In the United Kingdom, mothers who received counseling on exclusive breastfeeding (EBF) during prenatal care were 73% more likely to carry out EBF compared to those who did not. Similarly, mothers who received EBF counseling after childbirth were twice as likely to practice EBF compared to those who did not receive guidance on the subject. The study also showed that women who were knowledgeable about the duration and benefits of breastfeeding had a higher prevalence of EBF compared to others [29].

Counseling for women in same-sex relationships remains quite limited. All pregnant women need to receive support during the prenatal period, childbirth, and throughout their breastfeeding journey. Research in the United States shows that these mothers often seek information from external sources, such as empirical reports, books, online groups, and websites [45]. In Africa, 80% of women who breastfed within 1 h of their child’s birth received assistance from a health professional, while 47% of women who did not breastfeed within that first hour did not receive assistance from a health professional [38]. Other factors that interfere with breastfeeding are the knowledge of health professionals, the support of institutions, and the implementation of policies such as the Baby-Friendly Hospital Initiative (BFHI) [23].

The slow adoption of the BFHI policy by hospitals in Abu Dhabi is related to factors external to the health system and, as a result, most mothers adopt mixed feeding strategies for their children [46]. Many American mothers obtained information about breastfeeding in the hospital after giving birth, as well as received practical assistance to show them how to hold the baby to feed him/her. Participants who placed importance on the benefits of breastfeeding for their infants’ health and development generally continued to breastfeed longer [20,26].

Women’s intention to breastfeed and high self-efficacy were strong predictors of positive breastfeeding outcomes [26,30,43]. Furthermore, the acquisition of knowledge and practical skills by mothers, fathers/partners, and family members during pregnancy and postpartum visits is considered a cornerstone of successful breastfeeding [4]. Prenatal education and counseling for adolescents may be more effective if they include family members and peer support for adolescents.

A home visit or outpatient appointment for adolescents in the initial days following hospital discharge, along with practical support, could enhance their ability to breastfeed. Scheduled home visits during the exclusive breastfeeding period, up to six months postpartum, enhance the self-confidence and satisfaction of breastfeeding mothers [35,38]. Additionally, it is essential to expand support for the transition back to school and to create more breastfeeding-friendly environments in schools. These measures have the potential to help adolescents who wish to breastfeed continue successfully [20].

In addition to support from health professionals, support from family, spouse, and employer are essential links to successful breastfeeding. This includes social, instrumental, and emotional support, access to reliable information, and resources for community-based breastfeeding promotion programs. On the other hand, myths among low-income African American mothers and their families need to be dispelled [25,41].

Support from peers, spouses, and family members helped women feel more confident about breastfeeding and contributed to reducing stress and strain at work and breastfeeding [16,22]. A study conducted with women in Hong Kong found that breastfeeding support from family and friends was positively associated with the success of exclusive breastfeeding for up to six months, even during the COVID-19 pandemic [5].

Other factors that facilitate mothers’ breastfeeding practices after returning to work include employment benefits such as paid maternity leave, shorter commute time, a breastfeeding-friendly workplace, and flexible working hours [16,32]. Furthermore, having a daycare center in the workplace and positive attitudes and personal experiences of breastfeeding among colleagues and managers are factors that encourage breastfeeding [15].

## 4. Discussion

### 4.1. Design

There are two main reasons for using mixed methods in research. The first is complementarity of using both quantitative and qualitative data to gain a more comprehensive understanding of the phenomenon being studied. The second reason is sequential, where one research approach lays the foundation for the questions of the other, helping to fill in the gaps regarding the phenomenon of interest [11,47].

Integration in mixed methods research is classified into four categories: (1) data fusion, when qualitative and quantitative data are analyzed separately, then the results from both arms are compared for differences and similarities in a convergent design; (2) data explication, when qualitative findings help to explain quantitative findings in an explanatory sequential design; (3) data construction, when qualitative results inform future quantitative research questions and enable the development of an exploratory sequential design; and (4) data incorporation, when qualitative findings can be incorporated and help explain quantitative findings in an intervention design [11,48].

In this review, most studies used the explanatory sequential approach, deepening the findings about breastfeeding rates and duration, barriers, and facilitators of breastfeeding. Although the academic community and policymakers are calling for a broader application of mixed methods research, suggesting the combined use of quantitative and qualitative methods, its application has been present in studies conducted in some countries, notably the United States [20,21,25,30,40,41,45].

On the other hand, this type of research method remained stable between 2011 and 2019, with an upward curve in 2020 and 2022, indicating that mixed methods research is gaining strength, acquiring more prominence and value, probably due to a reflection of a change in research culture, moving from a single-method tradition to a multiple-method tradition.

In this scoping review, 50% of the studies analyzed used the explanatory sequential approach as a research design. It is pointed out that the most common design used in mixed study is convergent, in which QUAN and QUAL data collections are complementary and can be conducted separately or concomitantly [49].

### 4.2. Breastfeeding Rates Across Countries

The results highlighted that the prevalence of breastfeeding varied across countries, but in general, breastfeeding rates were lower in American, Mexican, Nigerian, and Irish women [20,27,30,41,42,44] than in women from Thailand, South Africa, China, and Sweden [26,32,37,38,43,44]. While the rates exceed the recommendations set by the WHO and UNICEF, the data in this scoping review cannot accurately determine the prevalence of breastfeeding among women. This limitation arises from the varied outcome measures used across the studies and the methodological weaknesses present in the included research.

### 4.3. Barriers and Facilitators to Breastfeeding

Individual and social barriers to breastfeeding, including lack of knowledge, embarrassment, generalized exposure to infant formula, inadequate maternity care practices, lactation difficulties, concerns about insufficient milk production, employment, insufficient family and social support, and inadequate knowledge and support from professionals, were reported by women. Conversely, some studies addressed the importance of support from health professionals to help mothers increase the duration and exclusivity of breastfeeding [5,20,21,23,24,25,27,28,29,31,41,42].

Family-centered breastfeeding models, peer support groups, and technologies have been studied as possible strategies to help women achieve their breastfeeding goals [20,22,23,27,30,50]. About breastfeeding facilitators, the literature indicates that continuous rooming-in, allowing mother and baby to stay together 24 ha day, creates a climate that potentially supports the mother in learning to recognize and respond to the baby’s hunger signals and instills confidence in her ability to care for the baby [7,42,51].

To enhance breastfeeding support and increase the rates of exclusive breastfeeding, public policies should focus on measures that improve mothers’ access to qualified assistance. Effective strategies include the availability of breastfeeding counselors and certified lactation consultants, such as International Board Certified Lactation Consultants (IBCLCs), who can assist with complex challenges. Additionally, postnatal home visits should be implemented to provide ongoing support. It is also crucial to establish a policy that prohibits the provision of formulas during the first 30 days of breastfeeding across all health settings. These initiatives aim to reduce barriers, promote exclusive breastfeeding, and ultimately improve child health outcomes [52].

### 4.4. Limitations of the Studies Included in the Review

Regarding the limitations of the studies evaluated in this review, approximately 5 studies (13.9%) discussed the findings regarding sampling (N). Although they used a mixed methods design to strengthen the conclusions, the quantitative and qualitative sample sizes were relatively small, which limits the generalization of the data [4,25,27,28]. In addition, five studies (13.9%) addressed concerns related to recall bias in conducting the research [5,6,29,36,44]. Only two studies (5.6%) mentioned the limitations associated with a low response rate and insufficient number of responses to allow adequate statistical analysis [22,30].

Another limitation observed was the lack of clarity regarding the theory used to support the analysis of the findings. For Creswell and Creswell [11], a mixed methods design, within a theoretical perspective, should present a formalized set of assumptions to guide the approach and conduct of the research. Two studies used the theory of planned behavior to analyze their data [19,20]. According to Neifert [53], this theory has been used to develop behavior change interventions, focusing on understanding the factors that influence the intention to exclusively breastfeed. Such interventions seek to promote positive behavioral beliefs, correct misconceptions, and empower women.

In mixed methods research, a basic premise is the need for clear integration between quantitative and qualitative data. This integration can be achieved through a joint display matrix [14], which facilitates the clear visual presentation of the findings, allowing us to identify how they connect, compare, and complement each other.

Among the articles analyzed, only two adopted this strategy. Witten et al. [24] used a matrix to present factors that influence exclusive breastfeeding in a cohort of mothers with babies aged 6 to 24 weeks, organizing the data by themes and codes, highlighting the most frequent barriers and facilitators. Felder et al. [25] investigated 50 pregnant African American women during an obstetric–gynecological consultation (Phase I) and, through a three-phase integration, presented the six qualitative themes (Phase II) and the integrated data (Phase III), revealing relevant cultural considerations for future research. This approach is evidenced as a best practice in mixed methods research, as it offers greater clarity in the relationship between qualitative and quantitative data, while enabling a richer and more visually accessible analysis. On the other hand, the other articles analyzed did not adopt the joint display matrix, presenting the data separately. Such a choice can make it difficult to immediately identify the relationships between the two data sets, compromising, to some extent, clarity and integration, which are fundamental pillars in mixed methods research.

## 5. Limitations of the Review and Scope of Mixed Methods Research

The limitation of this review is that the scope of mixed methods research may be much broader than what was captured since we cannot exclude the possibility that we missed some mixed methods publications that might not have been clearly identified as mixed methods. Furthermore, systematic review studies were excluded, and only primary studies were included. This is because the scoping review primarily focuses on primary studies, as they provide original data necessary for analyzing trends and identifying gaps. Systematic reviews, on the other hand, synthesize primary studies and, when included, may lead to redundancy without offering new insights, as their purpose is to consolidate existing data.

## 6. Conclusions

In this scoping review, most studies used the explanatory sequential approach, deepening the findings on breastfeeding rates and duration, as well as barriers and facilitators of this process. The convergent design was also widely used, in which the collections of quantitative and qualitative data are complementary and can occur separately or simultaneously. One limitation identified was the lack of clarity regarding the theory used to support the analysis of the results. Only two studies applied the theory of planned behavior to investigate the factors that influence the intention to exclusively breastfeed.

Breastfeeding rates varied between countries, being lower in women from the United States, Mexico, Nigeria, and Ireland and higher in women from Thailand, South Africa, China, and Sweden. However, the data did not allow for an accurate determination of the prevalence of breastfeeding, due to the variability between the contexts analyzed, differences in measurement methods, and methodological weaknesses in the studies.

The main barriers identified include lack of knowledge, embarrassment, exposure to infant formula, inadequate maternal care practices, difficulties with lactation, concerns about milk production, employment-related issues, and lack of family and professional support. Among the facilitators, the continuous rooming-in model stands out, which allows mother and baby to be together 24 h a day, helping the mother to recognize the baby’s hunger cues and building her confidence. In addition, education and counseling during the prenatal, labor, and postpartum periods, the woman’s intention to breastfeed, and support from family, employer, spouse, and health professionals were also identified as important factors.

The review recognizes that the scope of mixed methods research may be broader than captured, as some studies may not have been clearly identified as mixed methods studies. In addition, systematic review studies were excluded, considering only primary studies.

Breastfeeding is a crucial public health issue for mothers and children. The findings of this study can offer valuable insights for researchers, policymakers, and program managers, enabling them to develop more effective strategies to promote and support breastfeeding.

## Figures and Tables

**Figure 1 healthcare-13-00746-f001:**
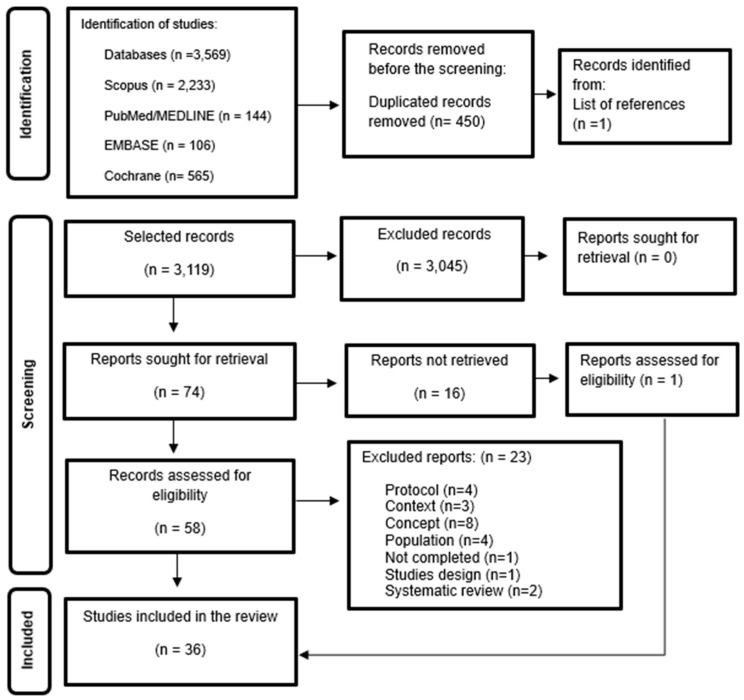
Studies’ Selection Flowchart—Prisma-SCR. Source: Peters et al. [13].

**Figure 2 healthcare-13-00746-f002:**
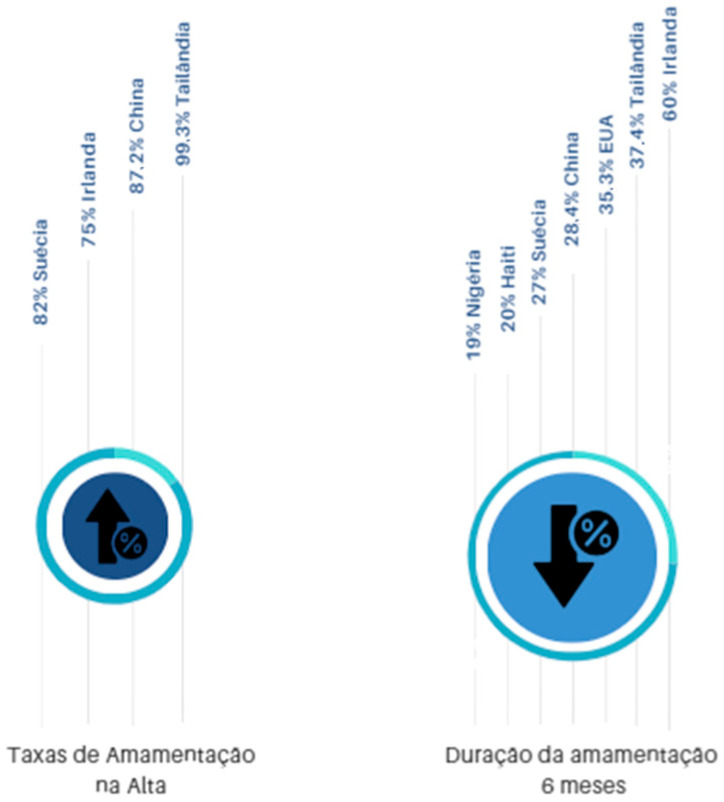
Breastfeeding rates and duration at hospital discharge and after 6 months of life. Source: Own authorship.

**Figure 3 healthcare-13-00746-f003:**
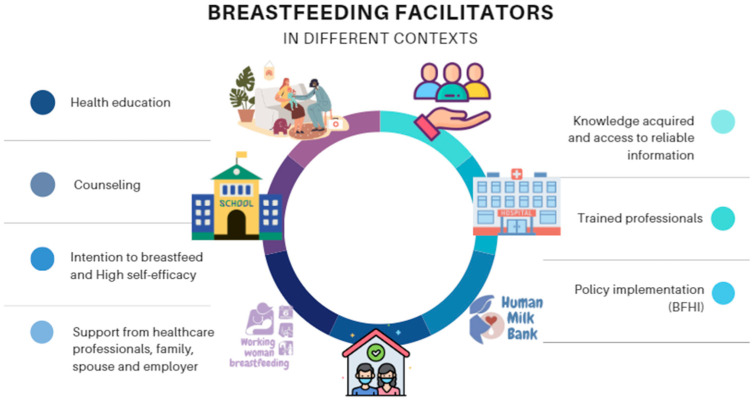
Breastfeeding facilitators in different contexts. Source: Own authorship.

**Table 1 healthcare-13-00746-t001:** Characterization of the articles included according to year and country of publication.

Variable		N° (Cases)	% (Approximate Value)
Year	2011	1	2.8%
	2012	2	5.6%
	2013	2	5.6%
	2015	2	5.6%
	2016	2	5.6%
	2017	1	2.8%
	2018	2	5.6%
	2019	3	8.2%
	2020	8	22.2%
	2021	4	11%
	2022	7	19.4%
	2023	2	5.6%
Country	United States	7	19.4%
	United Kingdom	2	5.6%
	Thailand	3	8.2%
	Australia	1	2.8%
	South Africa	1	2.8%
	Tanzania	1	2.8%
	Ghana	2	5.6%
	Sierra Leone	1	2.8%
	China	2	5.6%
	Mexico	2	5.6%
	Iran	1	2.8%
	Ireland	1	2.8%
	Iceland	2	5.6%
	Indonesia	2	5.6%
	Sweden	2	5.6%
	Nigeria	2	5.6%
	Denmark	1	2.8%
	Haiti	1	2.8%
	United Arab Emirates	2	5.6%

Note: The percentage values presented are approximate, based on the calculation of number of cases divided by 36, multiplied by 100. For illustration, 1 of 36 is approximately 2.8%, and 2 of 36 is approximately 5.6%. Source: Own authorship.

## Data Availability

All data analyzed during this study are included in this published article.

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
