# Peer review of "Mixed Methods Studies on Breastfeeding: A Scoping Review"

_healthcare, 2025, doi:10.3390/healthcare13070746_

Round 1
Reviewer 1 Report
Comments and Suggestions for Authors
The article needs to improve in several ways to achieve scientific writing. Enclosed are our comments in the attached document. the discussion part needs to be improved significantly. the introduction is still very modest; the author needs to provide the important and supportive evidence of the related topic in using the mixed-method research design.

Author Response
Dear Healthcare Reviewer,
Thank you for the opportunity to review our manuscript. We have addressed all the issues suggested by the reviewers, which has significantly improved our work. All changes are highlighted in the main text for your convenience.
Response to opinion
**Introduction:**
Comment 1: The authors are advised to revise a paragraph that contains at least three sentences. Please review this paragraph throughout the manuscript. The first paragraph does not adequately convey the quality of the research paper regarding the topic of breastfeeding. In the second paragraph, can the authors provide evidence of the potential success of this topic?
Answer 1: In response to these comments, we have revised the relevant paragraphs to provide a more detailed analysis of how mixed-methods research can enhance the understanding of breastfeeding. We have also included evidence demonstrating its effectiveness in this field. Our review clarifies how mixed methods contribute to understanding complex processes and the outcomes achieved through their application in breastfeeding studies.
Comment 2: We will also provide the initials of the co-authors in accordance with their roles: “Study Selection: Rayyan was utilized for selecting studies. Two researchers independently screened 109 studies based on titles and abstracts. A third reviewer resolved any conflicts regarding inclusion and exclusion. As per JBI recommendations, the flowchart model in Figure 1 has been used to detail the study selection process [11].”
Answer 2: This has been addressed and corrected.
Comment 3: **Eligibility Criteria:**
Answer 3: We have revised the paragraphs in the 'Method' section under 'Eligibility Criteria' to enhance clarity and precision.
Comment 4: **Discussion:**
Answer 4: We understand that the discussion should comprehensively address the substantive findings of each mixed-methods research (MMR) section. We have categorized some parts of the discussion into subheadings to improve readability and understanding.
Comment 5: Breastfeeding rates dropped within hours of birth: In Nigeria, the rate of mothers breastfeeding within the first hour of birth starts at 45% and decreases to 29% within the first two hours [25]. Another study in Nigeria reported that breastfeeding remained above 50% up to one hour after birth [4]. In Thailand, the cumulative proportions of mother-child pairs initiating breastfeeding within two, three, and four hours after birth were 92.2%, 92.4%, and 94.7%, respectively [24].
Answer 5: To conclude, we have added a summary at the end of the paragraph that discusses breastfeeding rates in the first few hours after birth. This conclusion emphasizes that, despite variations across countries, there is a general downward trend in breastfeeding rates during the initial hours after birth, with Thailand being an exception where rates remain high.
Comment 5: **Conclusion:**
Answer 6: We appreciate the suggestion for improving the conclusion of the article. As directed, we have restructured the conclusion to more clearly highlight the main findings of each mixed-methods study addressed in the paper. The revised wording aims to provide a more objective and concise overview, emphasizing the key conclusions from each approach and the most relevant findings from the analyses conducted. If there are any other suggestions or adjustments needed, we are available to assist you.
Kind regards.

Reviewer 2 Report
Comments and Suggestions for Authors
I congratulate you on conducting a comprehensive and detailed review of studies on breastfeeding using mixed methods. The study provides a valuable contribution to understanding the factors influencing breastfeeding and identifying barriers and facilitators in various socio-cultural contexts. However, there are some aspects that could improve the clarity, methodological rigor, and impact of the results on public health policies.
It would be beneficial to explicitly explain the reasons for excluding certain studies and to discuss the potential impact of this process on the generalizability of the conclusions.
Although the MMAT was used, the analysis does not clearly detail how the studies were classified based on methodological quality. A table or a graphical representation of the MMAT scores would enhance the transparency of the quality assessment.
It would also be useful to clearly state how possible methodological limitations (e.g., study selection, variation in definitions of exclusive breastfeeding rates) impact the interpretation of the results.
The results provide a valuable perspective on breastfeeding, but discussing the implications for public policies could be improved by including concrete recommendations for enhancing breastfeeding support practices.
In some sections, the formulation of ideas is dense and could be clarified to improve the accessibility of the text.
Comments on the Quality of English LanguageThe manuscript is well-written, clear, and logically structured. However, in some sections, the formulation of ideas is dense, and the sentences could be simplified to improve clarity and accessibility.
A careful review of grammatical constructions and the use of academic terminology is recommended to ensure more fluent and precise expression.
Author Response
Dear Healthcare Reviewer,
Thank you for the opportunity to review our manuscript. We have addressed all the issues suggested by the reviewers, which has significantly improved our work. All changes are highlighted in the main text for your convenience.
Response to opinion
Comment 1: I congratulate you on conducting a comprehensive and detailed review of breastfeeding studies using mixed methods. The study makes a valuable contribution to understanding the factors that influence breastfeeding and identifying barriers and facilitators in various sociocultural contexts. However, some aspects could be improved to enhance clarity, methodological rigor, and the impact of the results on public health policies.
It would be beneficial to explicitly explain the reasons for excluding certain studies and discuss the potential impact of this process on the generalizability of your conclusions.
Answer 1: I would like to note that the justification for excluding systematic review studies in a scoping review is addressed in the section titled "Limitations of the Review and Scope of Mixed Methods Research."
Comment 2: Although the Mixed Methods Appraisal Tool (MMAT) was used, the analysis does not clearly detail how the studies were scored based on methodological quality. Including a table or graphical representation of the MMAT scores would increase the transparency of the quality assessment.
Answer 2: We are very grateful for your service as a reviewer and for your valuable observations and suggestions; your expertise has been fundamental in improving the work. As requested, we have included the MMAT in a table format, representing the MMAT criteria. This adaptation aims to enhance the transparency of the manuscript evaluation process, making the analysis clearer and more objective. Additionally, the graph has been included in the supplementary file. Thank you again for your time and dedication!
Comment 3: It would also be helpful to clearly indicate how possible methodological limitations (e.g., study selection and variations in definitions of exclusive breastfeeding rates) affect the interpretation of results.
Answer 3: I would like to reiterate that the justification for excluding systematic review studies is detailed in the section "Limitations of the Review and Scope of Mixed Methods Research." The exclusion is based on several main reasons. First, scoping reviews primarily focus on primary studies, as they provide original data needed for trend analysis and gap identification. Systematic reviews are syntheses of primary studies, and including them could result in redundant information without adding new insights, as their main objective is to consolidate existing data.
Thank you for your understanding.
Comment 4: The results provide valuable insights into breastfeeding, but discussing their implications for public policy could be improved by including concrete recommendations to enhance breastfeeding support practices.
Answer 4: In response to these suggestions, we have added a section discussing the implications for public policy. Thank you again for your careful observation.
Comment 5: In some sections, the formulation of ideas is dense and could be clarified to improve the accessibility of the text.
Answer 5: Thank you for your suggestion. We have made the necessary revisions to simplify the sentences and enhance the clarity and accessibility of the text. We have restructured some passages to make them more direct and easier to understand, without compromising the content and accuracy of the information presented. We believe that these modifications better meet your recommendations and facilitate understanding of the role of mixed methods research in the study of breastfeeding.
**Comments on the Quality of the English Language:**
The manuscript is well-written, clear, and logically structured. However, in some sections, the formulation of ideas is dense, and sentences could be simplified to improve clarity and accessibility. A careful review of grammatical constructions and the use of academic terminology is recommended to ensure a more fluent and precise expression.

Reviewer 3 Report
Comments and Suggestions for Authors
In this study the authors focused on "Mixed methods studies on breastfeeding: a scope review", however, please consider my recommendations to improve the manuscript.
Introduction: It is suggested that authors add a paragraph on the importance of breastfeeding and its relation to communicable and non-communicable diseases to understand the importance of the topic.
Methodology: The methodology section was appropriate, comprehensive and well described. No changes are required.
Results:
1. Breastfeeding Rates and Duration: It is recommended to add a table to present a clear picture of the results for reader interest and understanding.
2. Barriers to breastfeeding: include few subheading describing the different barriers. It will help readers to evaluate the major factors and their impact on breastfeeding.
3. Facilitators of breastfeeding: please add a figure to clearly depict the facilitator of breastfeeding in different settings.
Limitation section: it should be place after discussion before conclusion section
Discussion section is well written and no changes are required, however, it would be better if authors add few lines about the importance of the breastfeeding.
Author Response
Dear Healthcare Reviewer,
Thank you for the opportunity to review our manuscript. We have addressed all the issues suggested by the reviewers, which has significantly improved our work. All changes are highlighted in the main text for your convenience.
Response to opinion
In this study, the authors focused on "Mixed Methods Studies on Breastfeeding: A Scoping Review." However, please consider my recommendations for improving the manuscript.
Comment 1: **Introduction:** I suggest that the authors include a paragraph discussing the importance of breastfeeding and its relationship with both communicable and non-communicable diseases. This would help underscore the significance of the topic.
Answer 1: Thank you for your suggestion. In response, we have addressed the impact of communicable and non-communicable diseases, as well as broader aspects of breastfeeding, in the Introduction section. We believe that this approach provides a comprehensive context highlighting the benefits of breastfeeding, aligning with the study's objective of exploring its various impacts. By integrating these health issues, we aim to present a more complete and well-founded view of the topic, as you suggested.
**Methodology:** The methodology section is appropriate, comprehensive, and well-described. No changes are necessary.
**Results:**
Comment 2: 1. **Breastfeeding Rates and Duration:** I recommend adding a table to present the results more clearly for the reader's interest and understanding.
Answer 2: - Done.
Comment 3: 2. **Barriers to Breastfeeding:** Please include a few subheadings to describe the different barriers. This will help readers better assess the main factors and their impact on breastfeeding.
Answer 3: - Subheadings detailing the various barriers to breastfeeding have been added to the manuscript, as per your recommendation. We greatly appreciate your valuable contributions, which have significantly enriched the content.
Comment 4: 3. **Breastfeeding Facilitators:** I suggest adding a figure to visually illustrate breastfeeding facilitators in different settings.
Answer 4: - We have included a figure that clearly illustrates breastfeeding facilitators in various contexts, aiming to make the description more visual and accessible.
Comment 5: **Limitations Section:** This section should be placed after the Discussion and before the Conclusion.
Answer 5: - We would like to inform you that the "Limitations" section has been relocated to follow the Discussion and precede the Conclusion, enhancing the organization and coherence of the text. We appreciate your feedback and are open to any further adjustments.
Comentário 6: **Discussão:** A seção de discussão está bem escrita e nenhuma alteração é necessária. No entanto, seria benéfico para os autores adicionar algumas linhas sobre a importância da amamentação.
Resposta 6: - Obrigado por este feedback e garantiremos a inclusão de conteúdo adicional destacando a importância da amamentação.

Reviewer 4 Report
Comments and Suggestions for Authors
Thank you for the opportunity to review your paper. Please, see below few comments you might consider or clarify:
In the results and discussion, you state that “breastfeeding rates in the selected studies remain lower than global recommendations.” However, given the heterogeneity of study designs, populations, and outcome measures, this statement risks oversimplifying the complex picture. Consider moderating this claim and discussing the variability across contexts rather than implying a uniform underperformance.
– In Table 1 (and any similar data summaries), the presentation of numbers (e.g., “1(2,8%)” vs. “2(5,6%)”) shows inconsistent use of decimal symbols (commas versus dots). Use a single style throughout (preferably a dot for decimal separation).
– Verify that the percentages correctly reflect the sample sizes (36 studies) and that rounding is consistent. For example, 1 out of 36 is approximately 2.8% and 2 out of 36 is approximately 5.6%, but ensure the table’s layout and column headings clearly explain these calculations.
Figure 1 (Flowchart):
– The flowchart is adapted from Peters et al. Make sure that all adaptations are clearly indicated and that the figure is self-contained (i.e., it includes a clear legend and appropriate labeling).
– Check that the figure numbering in the text matches the actual figures and that any supplementary figures (e.g., “Supl 2”) are properly referenced in the manuscript and provided in the supplementary file.
You report that only two studies met all MMAT criteria. It would strengthen the paper to explore what this implies about the broader literature. Is it an underestimation of methodological quality, or do the criteria highlight genuine deficiencies? Discuss whether the interpretation of these scores might have been influenced by different reporting standards across studies.
In several sections, the integration between the quantitative and qualitative findings is presented in a fragmented way. For example, when describing breastfeeding rates, barriers, and facilitators, the text sometimes appears to simply list numbers without synthesizing what the variation implies. Consider including a joint-display table or a more integrative narrative (even if most studies did not provide one) to clarify how the mixed methods components interrelate.
Best wishes
Author Response
Dear Healthcare Reviewer,
Thank you for the opportunity to review our manuscript. We have addressed all the
issues suggested by the reviewers, which has significantly improved our work. All
changes are highlighted in the main text for your convenience.
Response to opinion
Thank you for the opportunity to review your article. Below are some comments for
your consideration:
Comment 1: In your results and discussion, you state that “breastfeeding rates in
the selected studies remain lower than global recommendations.” However, due to
the heterogeneity of study designs, populations, and outcome measures, this
statement risks oversimplifying a complex issue. I suggest moderating this
statement and discussing the variability across different settings rather than
implying a uniform underperformance.
Answer 1: I would like to inform you that we have reviewed and modified the
statement regarding breastfeeding rates in the selected studies in accordance with
the feedback provided. The original phrase, "breastfeeding rates in the selected
studies remain lower than global recommendations," has been reformulated to be
more precise, considering the variability of the contexts analyzed.
These changes aim to enhance the clarity and coherence of the text while
acknowledging the diversity of study designs, population characteristics, and
methodologies that may influence the results. The modification also reflects a more
cautious approach when discussing disparities between observed rates and global
recommendations. We have fully accepted the suggestions, and the new wording
has already been incorporated into the text. I remain available for any additional
adjustments if necessary.
Comment 2: - In Table 1 (and other similar data summaries), there is an inconsistent
presentation of numbers (e.g., "1(2.8%)" vs. "2(5.6%)") due to the use of different
decimal symbols (commas vs. periods). Please use a consistent format (preferably
a period for decimal separation).
Answer 2: This has been addressed and corrected.
Comment 3: - Please ensure that the percentages accurately reflect the sample
sizes (36 studies) and that rounding is consistent. For instance, 1 in 36 is
approximately 2.8%, and 2 in 36 is approximately 5.6%. Make sure that the table
layout and column headings clearly explain these calculations.
Answer 3: We have made adjustments to the table and column headings to clarify
that the percentage calculations are approximate. An explanatory footnote has
been added to detail how the percentages were derived based on the number of
cases out of the total of 36. For example, 1 out of 36 corresponds to approximately
2.8%, and 2 out of 36 corresponds to approximately 5.6%. Additionally, we have
modified the column headings to indicate that the percentages are approximate. We
believe these changes improve the clarity and accuracy of the data presentation. I
am available for any questions or further adjustments as needed.
Comment 4: - The flowchart is based on Peters et al. Ensure that all adaptations are
clearly indicated and that the figure can stand alone (i.e., it includes a clear caption
and appropriate labeling).
Answer 4: There was an error in suggesting that the flowchart was "adapted" from
Peters et al. The authors intended to express that the flowchart was created based
on the data obtained. To avoid any misunderstanding, we have removed the term
"adapted" from the manuscript. We appreciate your attention to detail and are
available for any further clarification.
Comment 5: - Verify that figure numbering in the text corresponds with the actual
figures and that any supplemental figures (e.g., "Suppl 2") are properly referenced in
the manuscript and included in the supplemental file.
Answer 5: Figure numbering in the text has been corrected to align with the actual
figures. Additionally, supplementary figures (e.g., "Suppl 1") have been correctly
referenced in the manuscript and included in the supplemental file. We appreciate
your meticulousness and are open to making any further adjustments as necessary.
Thank you for your comment.
Comment 6: You report that only two studies met all the MMAT criteria. It would
strengthen the paper to explore what this implies about the wider literature. Is this
an underestimation of methodological quality, or do the criteria highlight genuine
shortcomings? Discuss whether the interpretation of these scores may have been
influenced by different reporting standards across studies.
Answer 6: I'd like to clarify that the statement regarding only two studies meeting all
the MMAT criteria was properly justified in the "Limitations of the Studies Included
in the Review" section. The lack of compliance with all the criteria was not
influenced by varying reporting standards among the studies, but rather by specific
issues related to the presentation of the data.
The criterion that most studies failed to meet was the joint display matrix, which
requires quantitative and qualitative data to be presented in an integrated manner.
This approach allows for a clearer and more comprehensive analysis of the
relationships between the two data sets. In the majority of the studies reviewed, the
data were presented separately, making it challenging to identify the interactions
and relationships between the quantitative and qualitative approaches.
This choice of presentation may compromise the clarity and integration of data,
which are essential aspects of mixed methods research. Consequently, this
limitation was highlighted in our review as a factor that may have hindered the full
utilization of the potential of mixed methods methodology.
Comment 7: In several sections, the integration of quantitative and qualitative
findings is presented in a fragmented manner. For example, when discussing
breastfeeding rates, barriers, and facilitators, the text occasionally lists numbers
without summarizing what the variations imply.
Answer 7: We would like to point out that a summary is provided at the end of some
paragraphs in the results section, particularly where breastfeeding rates are
discussed. This summary aims to offer a clearer and more objective view of the data,
enhancing the understanding of the results while maintaining the integrity of the
analysis.
We are available to discuss any other points you consider relevant.
Comment 8: Additionally, consider including a joint display table or a more
integrative narrative, even though most studies did not provide one, to clarify how
the mixed methods components interrelate.
Answer 8: Thank you for your comment. Regarding the joint display table, we would
like to clarify that it was not included because the analyzed articles did not present
data integration in a joint manner. Without explicit presentations of this integration
by the authors, it would not be appropriate to infer or extrapolate data that were not
clearly articulated in the articles. Our analysis strictly adhered to the information
provided by the authors to maintain the integrity of the results and avoid
inappropriate interpretations.

Round 2
Reviewer 1 Report
Comments and Suggestions for Authors
Thanks for the revised version of the manuscript. All comments and suggestions have been adressed very well. This version can be continued to the next stage.
Reviewer 3 Report
Comments and Suggestions for Authors
The authors have significantly improved the manuscript considering the suggestions and recommendations.
Reviewer 4 Report
Comments and Suggestions for Authors
Thank you for addressing the comments